# Antropo: An open-source platform to increase the anthropomorphism of the Franka Emika collaborative robot arm

**Constantin Scholz[1,2]\*, Hoang-Long Cao[1,3]\*, Ilias El Makrini[1,3], Bram Vanderborght[1,2]**

**1** Brubotics, Vrije Universiteit Brussel, Brussels, Belgium, **2** IMEC, Leuven, Belgium, **3** Flanders Make, Vrije Universiteit Brussel, Brussels, Belgium

\* constantin.florian.scholz@vub.be (CS); hoang.long.cao@vub.be (HLC)

## Abstract

Robot-to-human communication is important for mutual understanding during human-robot collaboration. Most of the current collaborative robots (cobots) are designed with low levels of anthropomorphism. Therefore, the ability of cobots to express human-like communication is limited. In this work, we present an open-source platform named Antropo to increase the level of anthropomorphism of Franka Emika—a widely used collaborative robot arm. The Antropo platform includes three modules: a camera module for expressing eye gaze, a light module for visual feedback, and a sound module for acoustic feedback. These modules can be rapidly prototyped through 3D printers, laser-cutters, and off-the-shelf components available at a low cost. The Antropo platform can be easily installed on the Franka Emika robot. The added communication channels can be synchronised with the robot's motions to enhance mutual understanding. All hardware CAD design files and software files are released. The platform can be used to study human-like behaviours of cobots and the effects of these behaviours on different aspects of human-robot collaboration. We demonstrate the Antropo platform in an assembly task in which the Franka Emika robot expresses various human-like communicative behaviours via the added communication channels. We also present two industrial applications in which the Antropo platform was customised for the Universal Robots UR16e.

## Introduction

Collaborative robots (cobots) are designed to work in close proximity collaboratively with human workers. In the ISO 8373:2012 standard, they are defined as robots designed for direct interaction with a human [1]. These robots are an essential component of the smart manufacturing revolution of Industry 4.0 and human-robot co-working in Industry 5.0 [2–4]. Traditionally, industrial robots usually operate at high speeds in caged environments far away from workers without mutual interaction not to pose a safety hazard [5]. On the contrary, collaborative robots are deployed in shared workspaces in which the robot and human are active at the same time and work alongside each other with high flexibility and dexterity [6, 7]. The tasks collaborative robots perform include those that are too mundane, dangerous, or

**Data Availability Statement:** All relevant data are within the paper and its Supporting information files.

**Funding:** The work leading to these results has received funding from the European Union's Horizon 2020 research and innovation program as part of the SOPHIA project under Grant Agreement No. 871237, the Flemish Government under the program "Onderzoeksprogramma Artificiële Intelligentie (AI) Vlaanderen", and the Horizon Europe Framework Programme as part of the euROBIN project under Grant Agreement No. 101070596. The funders had no role in study design, data collection and analysis, decision to publish, or preparation of the manuscript.

**Competing interests:** The authors have declared that no competing interests exist.

repetitive for the worker. Taking these tasks over from the human can increase productivity and workplace satisfaction for the worker [3, 8, 9]. With a worldwide structural change in employment due to an increase in automation and change in demographics and an ageing population [10], there will be an increase of robots utilised in industry and the need for the normalisation of close human-robot interaction. It will become essential for workers to accept, understand and enjoy the interactivity with collaborative robots.

While collaborative robots have certain advantages in the workplace, they might pose physical risks and cause psychological stress [11]. To ensure safe and effective collaboration, it is important to establish mutual understanding between humans and cobots to ensure safe, seamless and productive human-robot interaction [12]. Humans are known to use social cues to communicate, such as utterances, eye gaze, and body gestures. If collaborative robots are able to express similar social cues, the robot-to-human communication and mutual understanding will be improved. However, cobot manufacturers including Universal Robots, FANUC, KUKA, ABB, and Franka Emika commonly design collaborative robot arms with 6 or 7 degrees of freedom, smooth edges, and a lightweight design. Their "machine-like" appearance limits their ability to express social cues.

For cobots to express social cues, they should possess a higher degree of anthropomorphism, human-like qualities, either in their appearance or behaviour, or both [13]. Anthropomorphism is attributing human traits, emotions, or intentions to non-human entities [14]. It has been shown that the addition of anthropomorphic traits to robots increases the sympathy and willingness to work and interact with them [15]. Previous studies also have shown that robots with higher levels of anthropomorphism can increase people's perception of the robot's animacy, likability, intelligence, and enjoyment, as well as their intention to use the robot and their engagement with the task [16–18].

In this work, we propose Antropo, an open-source platform that enhances the anthropomorphism of the widely used Franka Emika collaborative robot arm, enabling it to express a more diverse range of social cues. The platform can be in the vicinity of the robot's end effector, known as the Franka Hand (see Fig 1) with three modules: a camera module to mimic eye gaze, a customisable light module for visual cues, and a sound module for auditory cues. The added visual and auditory communication channels allow the robot to convey a richer array of social cues and foster more natural interactions. It should be noted that our work adopts the common assumption in the social robotics field that social interactions with robots are due to anthropomorphism, while sociomorphism, or the perception of actual non-human social capacities, can also play a role [19].

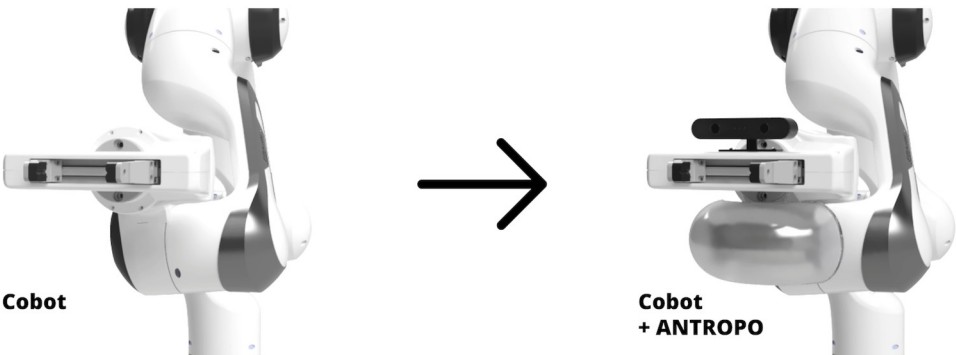

**Fig 1. Franka Emika Panda cobot without and with Antropo consisting of a camera, light and sound module.**

The Antropo platform's open-source design follows two key features of open-source hardware: the use of off-the-shelf components and the online availability of designs and materials. These features make the platform easy to construct and customise for users [20, 21]. We provide detailed assembly instructions, including all necessary hardware, software, and audio files, with the intention of enabling researchers to rapidly construct the platform and transfer the platform to other collaborative robot arms of different manufacturers. To demonstrate the platform's capabilities, we present two demonstrations conducted under EU-funded projects. In the first demonstration, we used the original Antropo platform with the Franka robot to investigate the effects of multi-modal social cues on human-robot collaboration. In the second demonstration, we used customised versions of the platform with the Universal Robots UR16e to explore the performance of a shared task between a human and a cobot in an industrial setting.

## Materials and methods

When designing the Antropo prototype, we followed the standard engineering design process including four steps and the two key features of open-source hardware: the use of off-the-shelf components and the online availability of designs and materials [20, 21].

1. **Problem Definition:** We identified the problem of limited robot-to-human communication in current collaborative robots and the need for increased anthropomorphism through anthropomorphic appearance and visual-auditory communication channels.

2. **Conceptual Design:** We generated the design concept for the Antropo platform, which includes three modules: a camera module for expressing eye gaze, a light module for visual feedback, and a sound module for acoustic feedback.

3. **Embodiment Design:** We developed the specifications of the Antropo platform, including its function, geometrics, physical compatibility with the Franka Emika cobo, and financial availability.

4. **Detailed Design:** We identified and established the properties of all the components inside the Antropo platform. We ensured that the platform is lightweight, non-intrusive, and easy to replicate. We also designed the platform to be rapidly prototyped through 3D printers, laser-cutters, and off-the-shelf components available at a low cost.

The following subsections present different modules of the Antropo platform and the reasons the design choices were made to increase the anthropomorphism of the Franka Emika Panda robot arm, see Fig 2. A complete User's Guide is included in the Supplementary Materials consisting of the design files, bill of materials, required tools, build instructions, and operation instructions. Future developments will be also updated at http://dx.doi.org/10.17632/9wnd37wv7c.1 [22]. Antropo holds an Open Source Hardware Association (OSHWA) license, bearing the unique identifier UID BE000008; OSHWA License: https://certification.oshwa.org/be000008.html, which demonstrates our commitment to the open-source community and fosters collaboration and innovation in the field of human-robot-interaction.

### Camera module

The camera module consists of a camera dummy that is screwed onto the end-effector to represent eyes and give the user the perception of the visual capabilities of the robot arm, inspired by [16]. Moreover, the camera looks similar to the camera used in collaborative robotics applications such as the Pickit L [23] or Intel RealSense [24] camera solutions. The reason for designing a camera dummy is to allow researchers to quickly study the robot's eye gaze without purchasing expensive equipment. Research results with this camera dummy can be transferable

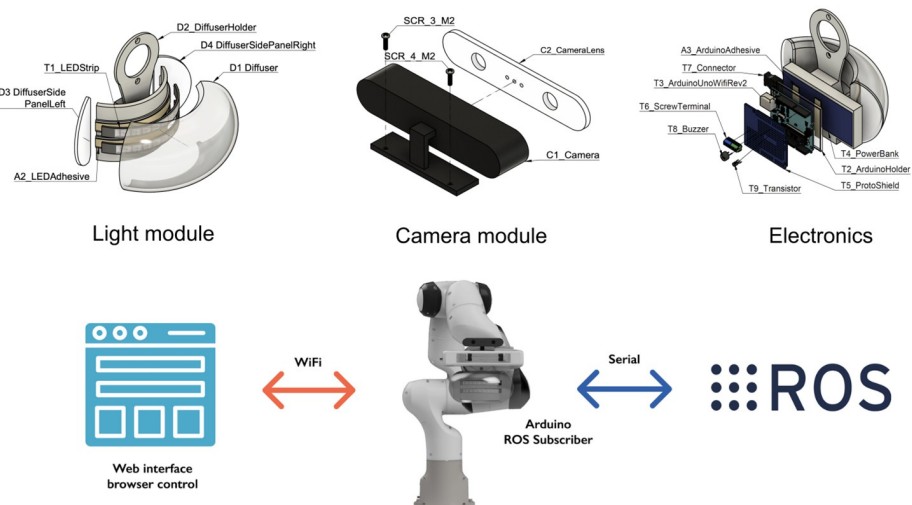

**Fig 2. Overview of the Antropo platform.** The platform can be controlled using a web interface or ROS.

onto applications where the before mentioned cameras are used. When looking frontally onto the gripper with the light module and camera installed with the gripper fingers being in the middle, it might remind the user of the movie character Wall-E designed by Disney Pixar [25].

## Light module

The light module is installed between the sixth link and the robot's end-effector (seventh link), compromising 28 RGB LEDs which are arranged in two strands under a translucent light diffuser. The light module is controllable via Wifi or ROS serial through an Arduino Uno Wifi Rev2 [26]. The location of the light module was selected to be placed on the robot to allow the incoming visual signals to be captured by the worker with ease using bottom-up attention and centralising the incoming sensory input [27]. The light module is designed to ensure the robot has an optimal spatial clearance for movement, even when engaging with large objects. Furthermore, its strategically positioned installation near the end-effector ensures it does not interfere with the robot's movements.

The design of the light module is inspired by the Philips Hue Flourish table light that compromises a contemporary and translucent spherical diffuser and allows remote colour control [28]. Choosing a design inspired by Philips Hue, and similar LED diffuser lamps allows us to build new interactions based on familiar forms that have been seen before by the worker in an everyday domain that will not feel foreign and allow intuitive understanding [29]. The design allows 180-degree visibility of the lights when facing the robot frontally. Finally, the light module is also designed to represent a chest or belly to give the robotic arm a higher level of anthropomorphism and capability, i.e. displaying breathing through dimming the LEDs in a breathing pattern [30].

We propose a selection of the light feedback cues for communicating the collaborative robot's intent which is predominately based on the human-machine interaction colour codes defined in IEC 60073:2002, see Fig 3 [31]. The international standard is based on standard colours for traffic lights and signs in road traffic. The cues presented here can be fully customised and supplemented if required for other experiments or interactions.

**White** The colour white indicates a neutral state and indicates the machine is active; the robot utilises this to communicate a standby or gazing state. To simulate the secondary action of

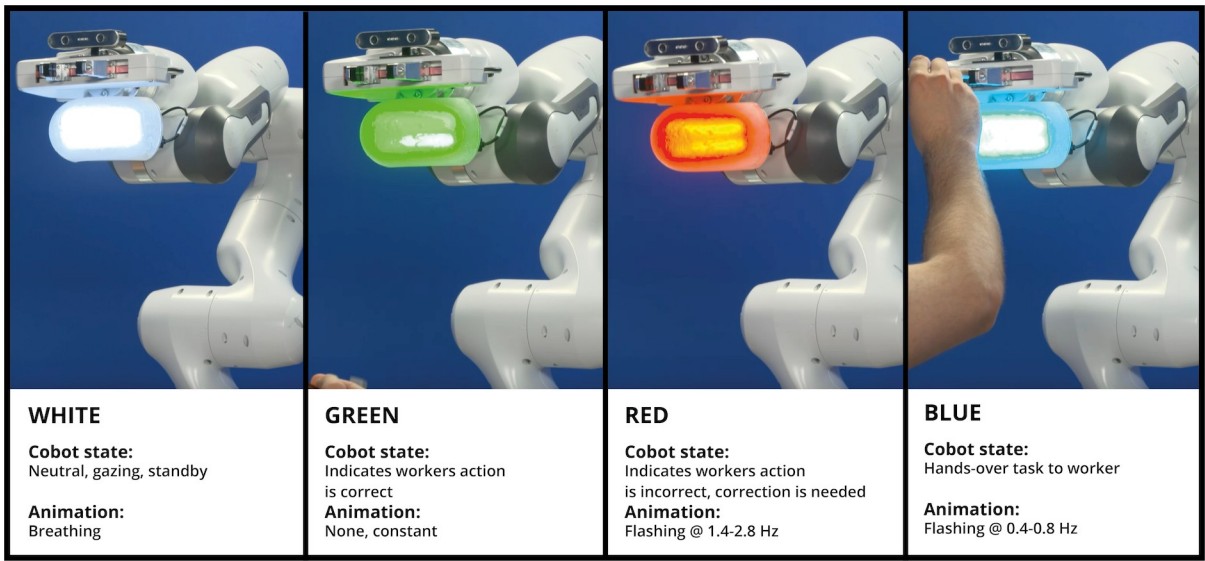

| **WHITE** | **GREEN** | **RED** | **BLUE** |
| --- | --- | --- | --- |
| **Cobot state:** Neutral, gazing, standby | **Cobot state:** Indicates workers action is correct | **Cobot state:** Indicates workers action is incorrect, correction is needed | **Cobot state:** Hands-over task to worker |
| **Animation:** Breathing | **Animation:** None, constant | **Animation:** Flashing @ 1.4-2.8 Hz | **Animation:** Flashing @ 0.4-0.8 Hz |

**Fig 3. Antropo's proposed colours for the light module in human-robot interaction.**

breathing, the lights are dimmed from full brightness to low brightness based on a healthy breathing pattern of 12–20 breaths per minute [32].

**Green** In the norm, the colour green means the correct and normal operation of a machine; the robot utilises this to communicate that the worker's action is correct.

**Red** The colour red indicates the stop of an operation and that immediate emergency action is required; the robot utilises this to communicate that the worker's current action is incorrect and needs correction. To increase the perceived importance, we decided to flash the red signal fast with 84–168 flashes per min (1.4 Hz to 2.8 Hz) as per the norm for high-priority attention attraction.

**Blue** The colour blue indicates that mandatory action or intervention is required; the robot utilises this to hand over the task for continuation to the worker. For the hand-over, we added a flashing of 24–48 (0.4 Hz to 0.8 Hz) flashes per minute, which is used for lower priority attention according to the norm.

## Sound module

The sound module consists of sound files that can add audio interaction cues to human-robot applications or experiments. To the best of our knowledge, we are the first to release and propose open-source sounds that can be used for human-robot interaction. We have composed three sounds inspired by the man-machine interaction acoustic codes defined in IEC 60073:2002 [31].

**S1: Hand-over sound** The hand-over sound is a single short high pitch tone. The sound can be used to indicate to the user that he needs to, i.e. take an object from the end effector.

**S2: Action correct sound** According to the norm, the correct sound falls under the category of other sounds and can be freely designed; it is inspired by the message sent tone utilised in iOS and is a rising tone [33]. The sound should be used when, i.e. the robot acknowledges a correct action of the worker.

**S3: Action incorrect sound** The incorrect sound is based on the norm for an emergency and utilises two bursts of sounds to indicate that, i.e. the worker has done an incorrect assembly step.

The sounds are played back with a piezoelectric buzzer. Additionally, all sounds can be downloaded as wav files from the repository and can be freely integrated into applications where audio feedback is required or should be explored further. We decided not to design a sound for the robot's neutral, gazing, or standby state since the norm states that no sound shall be applied for a normal safe condition.

## Results

We prototyped the Anthropo platform and the technical performance can be previewed at https://youtu.be/eAeAdPo-mJE. The platform has a total weight of 340g. This resulted in a reduction of the Franka Emika cobot's 3kg payload by around 11%. However, the platform's influence on the robot's operation is negligible when picking loads under 2.6kg. In the following subsections, we present two demonstrations of the Anthropo platform to highlight its potential applications in human-robot interaction research and industry.

### Using the original Antropo platform to investigate the effects of multi-modal social cues

The original Antropo platform with the Franka robot was used in a study to investigate the effects of multi-modal social cues in human-robot collaboration, see Fig 4 [34]. Via the Antropo platform, the Franka Emika robot could express different social cues with added visual-auditory communication channels. In two online experiments, participants rated Franka, when combined with the Antropo platform, significantly higher in six out of eight scales of perception and acceptance. Additionally, the understandability of the robot's social cues was also improved compared to when Franka was used alone. These results demonstrate that the Antropo platform enhances people's perception and acceptance of the robot.

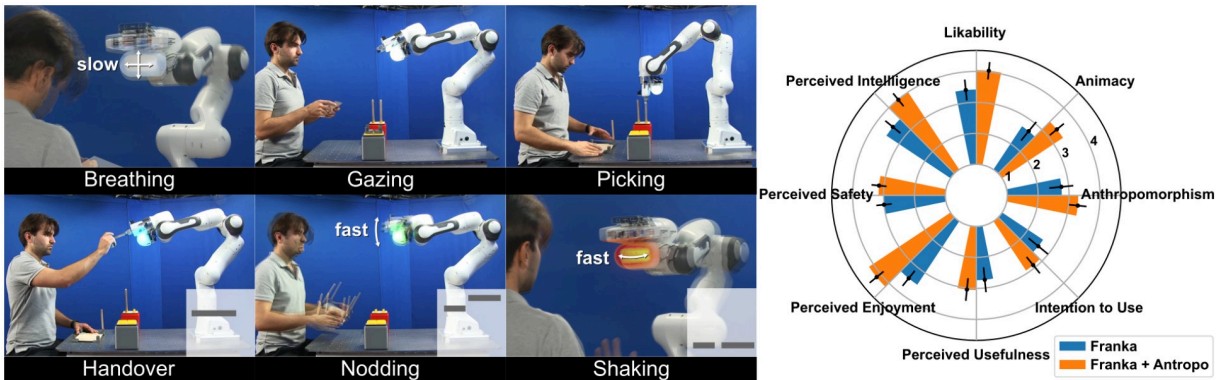

**Fig 4. The original design of Antropo.** (Left) Franka Emika robot with Antropo expresses different social cues. Melodies of the sound feedback are visualized at the bottom right of the images. (Right) Franka with the Antropo platform was rated higher in user perception and acceptance compared to no social cues. Error bars show the 95% confidence interval. The figure is similar but not identical to the original image used in the study [34], for illustrative purposes only.

### Using customised versions of Antropo with the Universal Robots UR16e

Two customised versions of the Antropo platform have been developed for the Universal Robots UR16s in two distinct industrial applications: gear deburring and assembly of an electric cabinet, see Fig 5. In both cases, the robot performed different movements, and the Antropo platform did not restrict any of the robot's movements. Preliminary analyses of interviews conducted with factory operators revealed that they found working with the system enjoyable because it provided them with more information about the robot's activities, and they expressed a willingness to utilise the robot with the Antropo platform.

## Discussion

We expect our platform to contribute to the research and development of collaborative robotics applications and the enhancement of human-robot interaction in industry and society. Researchers and users from industry can replicate the proposed platform design to increase the level of anthropomorphism of their Franaka Emika robot. Antropo can be used in both application and research contexts to improve the mutual understanding between humans and robots in industrial settings. For application purposes, the hardware can be implemented using related norms, standards, and findings from previous research on multi-modal social cues for collaborative robots. For research purposes, the hardware can be used to study two topics. The first topic is investigating the effects of multi-modal and non-verbal behaviours of collaborative robots on human workers or in broader contexts. The second topic is studying different designs of multi-modal social cues to improve mutual understanding. Furthermore, Antropo can be customised for other collaborative robot arms.

Since Antropo is integrated into a collaborative robot, safety considerations are essential when operating the Antropo platform. Antropo was specifically designed to ensure that it does not cause any movement limitations of the Franka Emika or result in self-collisions of the cobot and the platform. Notably, our implementation on the UR16 showcases the platform's capability to facilitate the manipulation of substantial objects without impeding the robot's kinematic performance. Nevertheless, it is practical to validate the programmed robot trajectory before execution to avoid unexpected collisions, particularly when the platform is customised with another Franka's end-effector or other collaborative robot arms such as different versions of the Universal Robots. Regarding external collision, the light module's resilient plastic shell shields the internal electronics, thereby minimising the probability of damage. In the event of shell breakage, the module can be easily replaced, preserving the platform's functionality while maintaining the integrity of other system components.

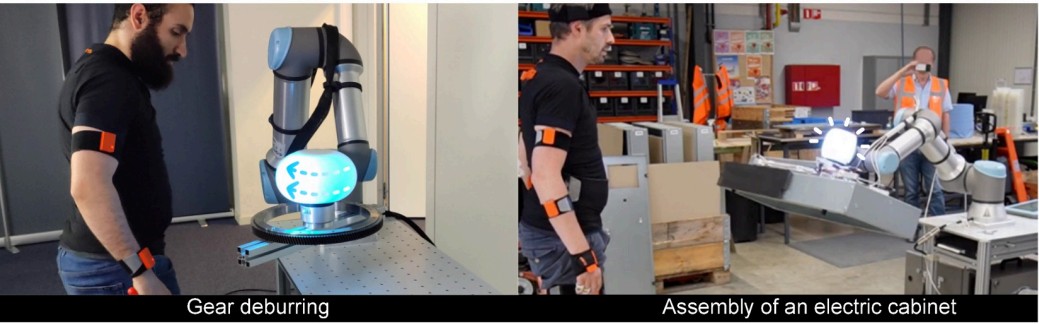

**Fig 5. Customised versions of Antropo with the Universal Robots UR16e: Gear deburring (left) and assembly of an electric cabinet (right).**

The selection of light colours for the Antropo platform was carefully considered and based on international standards that are commonly used in human-machine interaction and light traffic. These standards were appropriate for the Antropo platform as they provide clear and easily distinguishable colours that are suitable for conveying information to humans. However, as the level of anthropomorphism increases, the colours used in combination with the robot's embodiment can begin to take on additional meaning beyond their intended purpose. As can be seen in previous studies on social robotics, the use of certain colours in combination with the robot's behaviour can elicit emotions and potentially leads to emotional responses from humans [35–38]. Future research should aim to investigate this issue with the Antropo platform.

## Conclusion

This paper presents Antropo, an open-source platform designed to increase the anthropomorphism of the Franka Emika Panda robot arm. The platform utilizes commonly available off-the-shelf components and ensures simple, rapid manufacturing techniques. With added visual-auditory communication channels, Antropo enables the robot to express different social cues, enhancing mutual understanding between humans and robots in industrial settings. Antropo can be used for both application and research purposes, investigating the effects of multi-modal and non-verbal behaviors of collaborative robots on human workers and studying different designs of multi-modal social cues to improve mutual understanding. We anticipate that Antropo will contribute to the development of collaborative robotics applications and promote the enhancement of human-robot interaction in industry and society. Moreover, Antropo is expected to contribute to the growing body of knowledge surrounding the ethical implications of deploying anthropomorphic robots in the workplace.

## Supporting information

**S1 File. User's guide.** For design files, bill of materials, required tools, build instructions, and operation instructions.
(PDF)

**S1 Video. Video summary.** Antropo—Research platform demonstration.
(MP4)

## Acknowledgments

We would like to thank Lieven Bart Standaert (manager of FabLab Brussels) and Tom Turcksin (manager of AugmentX) for their advice and support with the Antropo project.

## Ethics statement

The individuals pictured in Figs 4 and 5 have provided written informed consent (as outlined in PLOS consent form) to publish their image alongside the manuscript.

## Author Contributions

**Conceptualization:** Constantin Scholz, Hoang-Long Cao, Ilias El Makrini, Bram Vanderborght.

**Data curation:** Constantin Scholz.

**Formal analysis:** Constantin Scholz.

**Funding acquisition:** Bram Vanderborght.

**Investigation:** Ilias El Makrini, Bram Vanderborght.

**Methodology:** Constantin Scholz, Hoang-Long Cao.

**Project administration:** Bram Vanderborght.

**Resources:** Constantin Scholz, Bram Vanderborght.

**Software:** Constantin Scholz.

**Supervision:** Ilias El Makrini, Bram Vanderborght.

**Validation:** Constantin Scholz, Hoang-Long Cao.

**Visualization:** Constantin Scholz, Hoang-Long Cao.

**Writing – original draft:** Constantin Scholz, Hoang-Long Cao, Ilias El Makrini, Bram Vanderborght.

**Writing – review & editing:** Constantin Scholz, Hoang-Long Cao, Ilias El Makrini, Bram Vanderborght.

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
