## [Decision Letter · Decision Letter 0]

20 Jun 2023

PONE-D-23-11948Antropo: An open-source platform to increase the anthropomorphism of the Franka Emika robotPLOS ONE

Dear Dr. Cao,

Thank you for submitting your manuscript to PLOS ONE. After careful consideration, we feel that it has merit but does not fully meet PLOS ONE’s publication criteria as it currently stands. Therefore, we invite you to submit a revised version of the manuscript that addresses the points raised during the review process.

We look forward to receiving your revised manuscript.

Kind regards,

Farzan Majeed Noori

Academic Editor

PLOS ONE

“The work leading to these results has received funding from the European Union's Horizon 2020 research and innovation program as part of the SOPHIA project under Grant Agreement No. 871237, the Flemish Government under the program "Onderzoeksprogramma Artificiële Intelligentie (AI) Vlaanderen", and euROBIN project. We would also like to thank Lieven Bart Standaert, manager of the FabLab Brussels, for his advice and support with the Antropo project.”

“The work leading to these results has received funding from the European Union’s 181 Horizon 2020 research and innovation program as part of the SOPHIA project under 182 Grant Agreement No. 871237, the Flemish Government under the program 183 “Onderzoeksprogramma Artifici¨ele Intelligentie (AI) Vlaanderen” and euROBIN project. 184 We would also like to thank Lieven Bart Standaert, manager of the FabLab Brussels, for 185 his advice and support with the Antropo project.”

“The work leading to these results has received funding from the European Union's Horizon 2020 research and innovation program as part of the SOPHIA project under Grant Agreement No. 871237, the Flemish Government under the program "Onderzoeksprogramma Artificiële Intelligentie (AI) Vlaanderen", and euROBIN project. We would also like to thank Lieven Bart Standaert, manager of the FabLab Brussels, for his advice and support with the Antropo project.”

7. We note that Figures 4a, 4b, 4c and 4d in your submission contain copyrighted images. All PLOS content is published under the Creative Commons Attribution License (CC BY 4.0), which means that the manuscript, images, and Supporting Information files will be freely available online, and any third party is permitted to access, download, copy, distribute, and use these materials in any way, even commercially, with proper attribution. For more information, see our copyright guidelines: http://journals.plos.org/plosone/s/licenses-and-copyright.

a. You may seek permission from the original copyright holder of Figures 4a, 4b, 4c and 4d to publish the content specifically under the CC BY 4.0 license.

b.If you are unable to obtain permission from the original copyright holder to publish these figures under the CC BY 4.0 license or if the copyright holder’s requirements are incompatible with the CC BY 4.0 license, please either i) remove the figure or ii) supply a replacement figure that complies with the CC BY 4.0 license. Please check copyright information on all replacement figures and update the figure caption with source information. If applicable, please specify in the figure caption text when a figure is similar but not identical to the original image and is therefore for illustrative purposes only.

Reviewers' comments:

Reviewer's Responses to Questions

**Comments to the Author**

1. Is the manuscript technically sound, and do the data support the conclusions?

Reviewer #1: Partly

Reviewer #2: No

2. Has the statistical analysis been performed appropriately and rigorously? 

Reviewer #1: N/A

Reviewer #2: N/A

3. Have the authors made all data underlying the findings in their manuscript fully available?

Reviewer #1: Yes

Reviewer #2: Yes

4. Is the manuscript presented in an intelligible fashion and written in standard English?

Reviewer #1: Yes

Reviewer #2: No

5. Review Comments to the Author

Reviewer #1: Summary

In this paper, the Antropo platform's open-source features are explained, which enable researchers and industry professionals to design anthropomorphic characteristics for cobots. By integrating cameras, light, and sound, the authors propose that cobots are better received by their users and enhance collaborations.

Strengths of the paper

• accessibility of the platform

• flexibility the system provides for both research and industry

• applicability in both industry and research

Suggestions for improvement

• Introduction: The introduction could be improved by adding a dedicated statement about the research aim and related work. Lots of work has been done in the social robotics field in regards to anthropomorphism, and the paper could benefit from adding some work from that field into this paper to make your arguments for specific features stronger, as now it seems a bit 'randomly' chosen.

• Materials: I wonder whether the placement of the features is not limiting the robot’s task performance or whether the system will not get damaged too easily as the gripper is the most used part of this robot. Perhaps it would be good to address this.

Furthermore, the choices for specific colors to indicate specific actions might not be as intuitive because certain colors also have an emotional connotation e.g. red being - angry and blue – sad. There is a paper by Johnson, D.O., Cuijpers, R.H., van der Pol, D.: Imitating human emotions with artificial facial expressions. International Journal of Social Robotics 5, 503–513 (2013) That looked at the meaning of different colored eyes for robots and how they are perceived. Perhaps that can help you to strengthen and or motivate your choices for colors of the eyes

• Results: it would be nice to implement some user evaluations of your setup to see how it is perceived to add more weight to the accuracy of the proposed actions. Even if it is just something very short as some interview of an exploratory survey.

Reviewer #2: Antropo: An open-source platform to increase the anthropomorphism of the Franka

Emika robot

This paper focuses on robot-to-human communication for human-robot collaboration. The paper indicates that currently there are a few collaborative robots (so called cobots) that are designed with anthropomorphic features. Further, the paper presents an open-source platform called Antropo that aims at increasing the anthropomorphism of Franka Emika cobot platform, through, for instance, added communication channels. The Antropo platform includes a camera module expressine eye gaze, a light module for visual feedback, and a sound module.

I will try to give the authors some input that can help them improve their paper. Please, see my comments below.

I would suggest that the title is adjusted to: “Antropo: An open-source platform to increase the anthropomorphism of the Franka Emika collaborative robotic arm”. In addition, that Franka Emika is a collaborative robotic arm should already be specified in the abstract.

In addition, collaborative robots/cobots should be clearly defined. Note, that the term “collaborative” is a positive value-ladden word that in some fields (e.g., Computer Supported Cooperative Work – see the work of Kjell Schmidt and Liam Bannon on collaboration vs. cooperation) indicates that a goal is achieved without conflicts, whereas the term “cooperation” (as opposed to “collaboration”) is a more neutral term, which does not imply that the goal is achieved in a friction-free way. To avoid confusion, indicate the definition of “collaborative robots” that you follow, and the corresponding reference.

Besides the “positive effects” of including a cobot in industry and/or in everyday life, please also specify eventual disadvantages (see the work from Martinetti et al. on safety, including the discussion on psychological safety: https://www.frontiersin.org/articles/10.3389/fceng.2021.666237/full).

Further, the authors argue for the anthropomorphization of these robotic arms: “ In this work, we aim to increase the level of anthropomorphism of the Franka Emika robot - a commonly used robot platform across research labs and seen as a reference platform for academic research due to its low-level access to robot control, and comparatively lower cost [13, 14].”

Please, bring more arguments on why these robots should be anthropomorphized. Be aware also of the critique from Johanna Seibt on the use of term “anthropomorphism” associated to robots. She and her colleagues argue rather for the use of term “sociomorphism”, where sociomoprhism can be seen at different levels. Please, check out her work: https://pure.au.dk/portal/en/publications/sociomorphing-not-anthropomorphizing(837953c2-3dc4-4507-9803-a3a3b6be3e8e).html.

Besides specifying the aim and bringing more arguments that support this aim, please, clearly specify the research question.

Regarding the method chosen, please, describe a scientific method (e.g., prototyping) rather than only describing the procedure and materials.

Regarding the results: was this platform/prototype tested with users? What do they say about the choice of colors, the use of camera in relation to their privacy, the choice of sounds etc.?

Note that the paper misses a discussion section. Please, include this in the paper.

6. PLOS authors have the option to publish the peer review history of their article (what does this mean?). If published, this will include your full peer review and any attached files.

Reviewer #1: No

Reviewer #2: No

<quillbot-extension-portal></quillbot-extension-portal>

---

## [Author Response · Author response to Decision Letter 0]

4 Aug 2023

Please also check our Response to Reviewers in PDF format..

Dear Editor and Reviewers,

Thank you for taking the time to review our manuscript. We greatly appreciate your recommendations concerning improvement to this paper. We have carefully considered your comments and have made significant revisions to address the issues you raised.

• We have formulated our research aim and validation with two demonstrations with industrial use cases.

• We have included more related studies to support the design of our open-source platform.

• We have mentioned the standard engineering design process in the Materials and Methods.

• We have added a Discussion section to discuss relevant issues suggested by the reviewers such as safetyand colours linked with emotion.

We believe that these changes have significantly improved the quality of the manuscript and we hope that you will find them satisfactory. We would be grateful if the revised version of our paper could be considered for publication in the journal.

Best regards,

Constantin Scholz, Hoang-Long Cao, Ilias El Makrini, and Bram Vanderborght

Reviewer #1

Comment: In this paper, the Antropo platform’s open-source features are explained, which enable researchers and industry professionals to design anthropomorphic characteristics for cobots. By integrating cameras, light, and sound, the authors propose that cobots are better received by their users and enhance collaborations.

Strengths of the paper: accessibility of the platform, flexibility the system provides for both research and industry, applicability in both industry and research.

Response: We appreciate your positive feedback on the accessibility and flexibility of the system and its applicability in both industry and research. We have read your recommendation carefully and revised the paper.

Comment: Introduction: The introduction could be improved by adding a dedicated statement about the research aim and related work. Lots of work has been done in the social robotics field in regards to anthropomorphism, and the paper could benefit from adding some work from that field into this paper to make your arguments for specific features stronger, as now it seems a bit ‘randomly’ chosen.

Response: We have added the research aim and more related work. More related studies have been included in the Introduction section.

"For cobots to express social cues, they should possess a higher degree of anthropomorphism, human-like qualities, either in their appearance or behaviour, or both (Duffy 2003). Anthropomorphism is attributing human traits, emotions, or intentions to non-human entities (Shin & Kim 2020). It has been shown that the addition of anthropomorphic traits to robots increases the sympathy and willingness to work and interact with them (Oistad et al. 2016). Previous studies also have shown that robots with higher levels of anthropomorphism can increase people’s perception of the robot’s animacy, likability, intelligence, and enjoyment, as well as their intention to use the robot and their engagement with the task (Sauer et al. 2021, Fischer et al. 2015, Faibish et al. 2022).""

The research aim has been formulated as follows.

"In this work, we propose Antropo, an open-source platform that enhances the anthropomorphism of the widely used Franka Emika collaborative robot arm, enabling it to express a more diverse range of social cues."

Comment: Materials: I wonder whether the placement of the features is not limiting the robot’s task performance or whether the system will not get damaged too easily as the gripper is the most used part of this robot. Perhaps it would be good to address this. Furthermore, the choices for specific colors to indicate specific actions might not be as intuitive because certain colors also have an emotional connotation e.g. red being - angry and blue – sad. There is a paper by Johnson, D.O., Cuijpers, R.H., van der Pol, D.: Imitating human emotions with artificial facial expressions. International Journal of Social Robotics 5, 503–513 (2013) That looked at the meaning of different colored eyes for robots and how they are perceived. Perhaps that can help you to strengthen and or motivate your choices for colors of the eyes

Response: Thank you for pointing out these issues. First, we added two industrial applications using the Antropo platform with different movements, which demonstrates that it does not limit task performance. Two customised versions of the Antropo platform have been developed for the Universal Robots UR16s in two distinct industrial applications: gear deburring and assembly of an electric cabinet, see Fig. 5. In both cases, the robot performed different movements, and the Antropo platform did not restrict any of the robot’s movements.

Second, we consider Antropo being damaged a low risk because the light module is made with a hard plastic shell that protects the electronics inside. In case the shell is broken, it can also be replaced easily without influencing other parts of the platform. We have added this point to the Discussion.

"Since Antropo is integrated into a collaborative robot, safety considerations are essential when operating the Antropo platform. Antropo was specifically designed to ensure that it does not cause any movement limitations of the Franka Emika or result in self-collisions of the cobot and the platform. Notably, our implementation on the UR16 showcases the platform’s capability to facilitate the manipulation of substantial objects without impeding the robot’s kinematic performance. Nevertheless, it is practical to validate the programmed robot trajectory before execution to avoid unexpected collisions, particularly when the platform is customised with another Franka’s end-effector or other collaborative robot arms such as different versions of the Universal Robots. Regarding external collision, the light module’s resilient plastic shell shields the internal electronics, thereby minimising the probability of damage. In the event of shell breakage, the module can be easily replaced, preserving the platform’s functionality while maintaining the integrity of other system components."

Third, our light colour selection is based on the colour codes defined in the international standard IEC 60073:2002 for human-machine interaction colour codes defined in IEC 60073:2002. These colours in this standard are also used for traffic lights and signs in road traffic. However, we agree that with a higher level of anthropomorphism, the colours can be linked to emotions. In the scope of this open-source design paper, we did not address this issue but in future studies. We have added this point to the Discussion.

"The selection of light colours for the Antropo platform was carefully considered and based on international standards that are commonly used in human-machine interaction and light traffic. These standards were appropriate for the Antropo platform as they provide clear and easily distinguishable colours that are suitable for conveying information to humans. However, as the level of anthropomorphism increases, the colours used in combination with the robot’s embodiment can begin to take on additional meaning beyond their intended purpose. As can be seen in previous studies on social robotics, the use of certain colours in combination with the robot’s behaviour can elicit emotions and potentially leads to emotional responses from humans (Haring¨ et al. 2011, Johnson et al. 2013, De Beir et al. 2016, Guo et al. 2019). Future research should aim to investigate this issue with the Antropo platform."

Comment: Results: it would be nice to implement some user evaluations of your setup to see how it is perceived to add more weight to the accuracy of the proposed actions. Even if it is just something very short as some interview of an exploratory survey.

Response: We have included two demonstrations from our ongoing projects. In the first demonstration, in an online study, we showed that Franka with the Antropo platform was rated higher in user perception and acceptance compared to no social cues (Cao et al. 2023).

In the second demonstration, Antropo was customised for the Universal Robots UR16e. Preliminary interview data showed that factory operators agreed that the Antropo platform helped them understand what the robot is doing, and they are willing to work with the robot with Antropo.

Reviewer #2

Comment: This paper focuses on robot-to-human communication for human-robot collaboration. The paper indicates that currently there are a few collaborative robots (so called cobots) that are designed with anthropomorphic features. Further, the paper presents an open-source platform called Antropo that aims at increasing the anthropomorphism of Franka Emika cobot platform, through, for instance, added communication channels. The Antropo platform includes a camera module expressine eye gaze, a light module for visual feedback, and a sound module.

I will try to give the authors some input that can help them improve their paper. Please, see my comments below.

Response: Thank you for your valuable feedback on our work. We have carefully considered your comments and have made revisions accordingly.

Comment: I would suggest that the title is adjusted to: “Antropo: An open-source platform to increase the anthropomorphism of the Franka Emika collaborative robotic arm”. In addition, that Franka Emika is a collaborative robotic arm should already be specified in the abstract.

Response: We have adjusted to title and the abstract based on your comment to specify Franka Emika collaborative robotic arm. New title: “Antropo: An open-source platform to increase the anthropomorphism of the Franka Emika collaborative robot arm”.

"In this work, we present an open-source platform named Antropo to increase the level of anthropomorphism of Franka Emika - a widely used collaborative robot arm."

Comment: In addition, collaborative robots/cobots should be clearly defined. Note, that the term “collaborative” is a positive value-ladden word that in some fields (e.g., Computer Supported Cooperative Work – see the work of Kjell Schmidt and Liam Bannon on collaboration vs. cooperation) indicates that a goal is achieved without conflicts, whereas the term “cooperation” (as opposed to “collaboration”) is a more neutral term, which does not imply that the goal is achieved in a friction-free way. To avoid confusion, indicate the definition of “collaborative robots” that you follow, and the corresponding reference.

Response: We agree that in other fields, the term “collaborative” might cause confusion. To avoid possible confusion suggested by the reviewer, we added a definition of “collaborative robot” according to the ISO 8373:2012 in the field of collaborative robotics.

"Collaborative robots (cobots) are designed to work in close proximity collaboratively with human workers. In the ISO 8373:2012 standard, they are defined as robots designed for direct interaction with a human (ISO 2012)."

Comment: Besides the “positive effects” of including a cobot in industry and/or in everyday life, please also specify eventual disadvantages (see the work from Martinetti et al. on safety, including the discussion on psychological safety: https://www.frontiersin.org/articles/10.3389/fceng.2021.666237/full).

Response: We have added the disadvantages of cobot deployments mentioned in Martinetti et al. (2021), which leads to the importance of mutual understanding. This is what we would like to achieve through the Antropo open-source platform.

"While collaborative robots have certain advantages in the workplace, they might pose physical risks and cause psychological stress (Martinetti et al. 2021). To ensure safe and effective collaboration, it is important to establish mutual understanding between humans and cobots to ensure safe, seamless and productive human-robot interaction (Sciutti et al. 2018)."

Comment: Further, the authors argue for the anthropomorphization of these robotic arms: “ In this work, we aim to increase the level of anthropomorphism of the Franka Emika robot - a commonly used robot platform across research labs and seen as a reference platform for academic research due to its low-level access to robot control, and comparatively lower cost [13, 14].”

Please, bring more arguments on why these robots should be anthropomorphized. Be aware also of the critique from Johanna Seibt on the use of term “anthropomorphism” associated to robots. She and her colleagues argue rather for the use of term “sociomorphism”, where sociomoprhism can be seen at different levels. Please, check out her work: https://pure.au.dk/portal/en/publications/ sociomorphing-not-anthropomorphizing(837953c2-3dc4-4507-9803-a3a3b6be3e8e).html.

Response: Thank you for providing a very interesting paper and relevant to our work. We used “anthropomorphism” because it is a widely used concept in the field. We added our recommendation of sociomoprhism vs anthropomorphism to improve the Introduction part of our paper.

"It should be noted that we adopted the common assumption in the social robotics field that social interactions with robots are due to anthropomorphism while sociomorphism, the perception of actual non-human social capacities, can also play a role (Seibt et al. 2020)."

Comment: Besides specifying the aim and bringing more arguments that support this aim, please, clearly specify the research question.

Response: Since our work is a open-source design paper, we formulate the research aim as follows.

"In this work, we propose Antropo, an open-source platform that enhances the anthropomorphic capabilities of the widely used Franka Emika collaborative robot arm, enabling it to express a more diverse range of social cues.

We demonstrate how we reached the aim by the preliminary results of two experiments.

"To demonstrate the platform’s capabilities, we present two experiments conducted under EU-funded projects. In the first experiment, we used the original Antropo platform with the Franka robot to investigate the effects of multi-modal social cues on human-robot collaboration. In the second experiment, we used a customised version of the platform with the Universal Robots UR16e to explore the performance of a shared task between a human and a cobot in an industrial setting."

Comment: Regarding the method chosen, please, describe a scientific method (e.g., prototyping) rather than only describing the procedure and materials.

Response: We have added the standard engineering design process in the Materials and Methods, which we followed during the design process.

"When designing the Antropo prototype, we followed the standard engineering design process including four steps and the two key features of open-source hardware: the use of off-the-shelf components and the online availability of designs and materials (Maia Chagas 2018, Bonvoisin et al. 2017).

1. Problem Definition: We identified the problem of limited robot-to-human communication in current collaborative robots and the need for increased anthropomorphism through anthropomorphic appearance and visual-auditory communication channels.

2. Conceptual Design: We generated the design concept for the Antropo platform, which includes three modules: a camera module for expressing eye gaze, a light module for visual feedback, and a sound module for acoustic feedback.

3. Embodiment Design: We developed the specifications of the Antropo platform, including its function, geometrics, physical compatibility with the Franka Emika cobo, and financial availability.

4. Detailed Design: We identified and established the properties of all the components inside the Antropo platform. We ensured that the platform is lightweight, non-intrusive, and easy to replicate. We also designed the platform to be rapidly prototyped through 3D printers, laser-cutters, and off-the-shelf components available at a low cost."

Comment: Regarding the results: was this platform/prototype tested with users? What do they say about the choice of colors, the use of camera in relation to their privacy, the choice of sounds etc.?

Response: We have tested the Antropo platform and in the revised version, we present two demonstrations from our ongoing projects. In the first demonstration, in an online study, we showed that Franka with the Antropo platform was rated higher in user perception and understanding of social cues due to the light and sound modalities (Cao et al. 2023).

In the second demonstration, Antropo was customised for the Universal Robots UR16e. Preliminary interview data showed that factory operators agreed that the Antropo platform helped them understand what the robot is doing, and they are willing to work with the robot with Antropo.

The open-source platform also allows researchers to investigate different light colors and sounds in combination with robot actions. Regarding the use of camera and privacy, we, unfortunately, did not investigate in the open-source platform paper.

Comment: Note that the paper misses a discussion section. Please, include this in the paper.

Response: We mixed the results and discussion in the original manuscript. In the revised version, we have separated these sections. Moreover, we have discussed more relevant issues recommended by both reviewers.

References

Bonvoisin, J., Mies, R., Boujut, J.-F. & Stark, R. (2017), ‘What is the “source” of open source hardware?’, Journal of Open Hardware 1(1).

Cao, H.-L., Scholz, C., De Winter, J., Makrini, I. E. & Vanderborght, B. (2023), ‘Investigating the role of multi-modal social cues in human-robot collaboration in industrial settings’, International Journal of Social Robotics pp. 1–11.

De Beir, A., Cao, H.-L., Gomez Esteban, P., Van de Perre, G., Lefeber, D. & Vanderborght, B. (2016), ‘Enhancing´ emotional facial expressiveness on nao: A case study using pluggable eyebrows’, International Journal of Social Robotics 8, 513–521.

Duffy, B. R. (2003), ‘Anthropomorphism and the social robot’, Robotics and autonomous systems 42(3-4), 177–190.

Faibish, T., Kshirsagar, A., Hoffman, G. & Edan, Y. (2022), ‘Human preferences for robot eye gaze in human-to-robot handovers’, International Journal of Social Robotics 14(4), 995–1012.

Fischer, K., Jensen, L. C., Kirstein, F., Stabinger, S., Erkent, O., Shukla, D. & Piater, J. (2015), The effects of¨ social gaze in human-robot collaborative assembly, in ‘Social Robotics: 7th International Conference, ICSR 2015, Paris, France, October 26-30, 2015, Proceedings 7’, Springer, pp. 204–213.

Guo, F., Li, M., Qu, Q. & Duffy, V. G. (2019), ‘The effect of a humanoid robot’s emotional behaviors on users’ emotional responses: Evidence from pupillometry and electroencephalography measures’, International Journal of Human–Computer Interaction 35(20), 1947–1959.

Haring, M., Bee, N. & Andr¨ e, E. (2011), Creation and evaluation of emotion expression´ with body movement, sound and eye color for humanoid robots, in ‘2011 RO-MAN’, IEEE, pp. 204–209.

ISO (2012), ‘Iso 8373: 2012 (en), robots and robotic devices—vocabulary’.

Johnson, D. O., Cuijpers, R. H. & van der Pol, D. (2013), ‘Imitating human emotions with artificial facial expressions’, International Journal of Social Robotics 5, 503–513.

Maia Chagas, A. (2018), ‘Haves and have nots must find a better way: The case for open scientific hardware’, PLoS biology 16(9), e3000014.

Martinetti, A., Chemweno, P. K., Nizamis, K. & Fosch-Villaronga, E. (2021), ‘Redefining safety in light of human-robot interaction: A critical review of current standards and regulations’, Frontiers in chemical engineering 3, 32.

Oistad, B. C., Sembroski, C. E., Gates, K. A., Krupp, M. M., Fraune, M. R. & Sabanoviˇ c, S. (2016), Colleague or´ tool? interactivity increases positive perceptions of and willingness to interact with a robotic co-worker, Lecture Notes in Computer Science, Springer International Publishing, pp. 774–785. Book Title: Social Robotics ISSN: 0302-9743.

Sauer, V., Sauer, A. & Mertens, A. (2021), ‘Zoomorphic gestures for communicating cobot states’, IEEE Robotics and Automation Letters 6(2), 2179–2185. URL: http://arxiv.org/abs/2102.10825

Sciutti, A., Mara, M., Tagliasco, V. & Sandini, G. (2018), ‘Humanizing human-robot interaction: On the importance of mutual understanding’, IEEE Technology and Society Magazine 37(1), 22–29. Conference Name: IEEE Technology and Society Magazine.

Seibt, J., Vestergaard, C. & Damholdt, M. F. (2020), ‘Sociomorphing, not anthropomorphizing: towards a typology of experienced sociality’, Culturally Sustainable Social Robotics–Proceedings of Robophilosophy pp. 51–67.

Shin, H. I. & Kim, J. (2020), ‘My computer is more thoughtful than you: Loneliness, anthropomorphism and dehumanization’, Current Psychology 39(2), 445–453. URL: http://link.springer.com/10.1007/s12144-018-9975-7

---

## [Editor Report · Decision Letter 1]

12 Sep 2023

Antropo: An open-source platform to increase the anthropomorphism of the Franka Emika collaborative robot arm

PONE-D-23-11948R1

Dear Dr. Cao,

We’re pleased to inform you that your manuscript has been judged scientifically suitable for publication and will be formally accepted for publication once it meets all outstanding technical requirements.

Kind regards,

Farzan Majeed Noori

Academic Editor

PLOS ONE

Additional Editor Comments (optional):

The paper has been revised as per authors suggestions and comments, I am happy to accept it in the latest form, Congratulations authors!
---

## [Editor Report · Acceptance letter]

25 Sep 2023

PONE-D-23-11948R1 

Antropo: An open-source platform to increase the anthropomorphism of the Franka Emika collaborative robot arm 

Dear Dr. Cao:

I'm pleased to inform you that your manuscript has been deemed suitable for publication in PLOS ONE. Congratulations! Your manuscript is now with our production department. 

Kind regards, 

on behalf of

Dr Farzan Majeed Noori 

Academic Editor

PLOS ONE